# Very Large Pore Mesoporous Bioactive Silicate Glasses: Comparison of Behavior toward Classical Mesoporous Bioactive Glasses in Terms of Drug Loading/Release and Bioactivity

**DOI:** 10.3390/ma17020373

**Published:** 2024-01-11

**Authors:** Debora Carrozza, Erika Ferrari, Gianluca Malavasi

**Affiliations:** Department of Chemical and Geological Sciences, University of Modena and Reggio Emilia, Via G. Campi 103, 41125 Modena, Italy; debora.carrozza@unimore.it (D.C.); erika.ferrari@unimore.it (E.F.)

**Keywords:** large pore mesoporous silica, nisin, bioactive glasses, Ga-containing glasses, pharmaceutical peptides, porous biomaterials

## Abstract

Considering the increase in patients who suffer from osteoporosis and the bone defects that occur in these patients, bone tissue regeneration is a promising option to solve this problem. To achieve a synergistic effect between the synthesis of a proper structure and bioactive/pharmaceutical activity, ions with a physiological effect can be added to silica structures, such as Ca^2+^, thanks to its bioactive behavior, and Ga^3+^ for its antibacterial and anticancer action. In this work, the synthesis of large pore mesoporous silica (LPMS), potential bioactive glasses containing Ca^2+^ and Ga^3+^, has been studied. Corresponding structures, in terms of composition, have been synthesized following the Sol-Gel EISA (Evaporation Induced Self-Assembly) process (obtaining Classical Mesoporous Silica, MS). Pore structure characterization of LPMSs and MSs has been performed using N_2_ adsorption/desorption and Hg-porosimetry, showing the presence of pores for LPMSs in the range of 20–60 and 200–600 nm. Nisin, a polycyclic antibacterial peptide, has been used for load tests. The load and release tests performed highlight a higher loading and releasing, doubled for LPMSs if compared to MSs. To confirm the maintenance of the structure of LPMSs and their mechanical strength and resistance, scanning electron microscopy images were acquired before and after release tests. Ca and Ga release in SBF has been studied through inductively coupled plasma—optical emission spectroscopy (ICP-OES), showing a particularly high release of these ions performed with LPMSs. The bioactive behavior of Ca-containing structures has been confirmed using FT-IR (Fourier-transform infrared spectroscopy), SEM-EDS (Scanning Electron Microscope-Energy Dispersive Spectroscopy), and X-ray powder diffraction (XRDP). In conclusion, LPMSs showed better loading and releasing properties compared with classical MS and better release in terms of active ions. In addition, it has also been demonstrated that LPMSs have bioactive behavior (a well-known characteristic of MSs).

## 1. Introduction

Big molecules with a pharmaceutical role, such as peptides or proteins, are becoming increasingly important in the last few years. Synthesizing a structure that can accommodate them is important to shield, stabilize, protect, and maintain their biological activity. To achieve this purpose, large pore mesoporous silica (LPMS) is an innovative support [1].

The prevalence of osteoporosis is on the rise as the population ages rapidly [2]. Healing bone defects in individuals with osteoporosis poses a challenge, attributed to the diminished capacity for bone formation and heightened bone resorption activity in the affected zones [3]. Bone diseases are one of the main challenges facing today’s society since bone is the second most transplanted tissue in the world [4]. Bone tissue regeneration has emerged as a promising option to solve this problem [5,6]. In this view, it is possible to produce materials able to replace, as a temporary support, a part of a bone and induce cell proliferation and cell differentiation [7].

In this sense, mesoporous bioactive glasses are going to play an important role in bone regeneration because of their outstanding textural properties, quick bioactive response, and biocompatibility [8]. 

Mesoporous bioactive glasses can release the Si^4+^ ions that benefit the presence of extracellular events, including angiogenesis, and the Ca^2+^ ions that contribute to cell proliferation and exhibit osteogenic activity, as well as play a very important role in gene transfection [9,10].

In recent years, mesoporous bioactive glasses for clinical applications have evolved from an improved bioactive biomaterial towards a multitherapy system for the treatment of different bone-related pathologies by tuning their composition and functionalizing their surfaces [10].

With this challenging perspective, mesoporous bioactive glasses have been developed and enhanced by the addition of biological functions through the inclusion of controlled amounts of therapeutic elements (Cu, Zn, Ga, Sr, Li, Ce, etc.) while maintaining the necessary bioactivity and biocompatibility and a highly exposed surface area [7].

The conventional approach for treating osteoporotic bone defects involves bone grafting combined with the systemic administration of anti-osteoporosis drugs. Nevertheless, systemic administration encounters challenges such as low availability, liver damage, and osteonecrosis of the jaw [11]. There is an anticipation that bone grafts, designed to enhance osteoblast activities and inhibit osteoclast activities, will prove effective in efficiently regenerating osteoporotic bone defects [12,13].

Calcium silicate (CaSiO_3_) possesses outstanding biological functions for bone regeneration [13,14], and this effect can be synergistically coupled with the effect of therapeutic metals such as gallium, which can suppress osteoclast activities [12,13,14,15,16,17] and is a conventional chemotherapeutic drug [15]. 

The addition of gallium to bioactive materials has been documented to boost osteogenesis, impact blood clotting, and elicit anticancer and antibacterial effects. The crucial discovery is that incorporating gallium into biomaterials holds significant promise for addressing bone-related diseases, given its efficient and controllable transfer to the targeted region [18].

Exploration of gallium’s potential anticancer properties was prompted by the observation that ^67^Ga, when administered to rodents with implanted tumors, exhibited a notable concentration within the tumors [19]. Consequently, initially, ^67^Ga was employed as a probe for tumor detection.

Gallium exhibits the capability to mimic iron and disrupt iron-dependent processes, including proliferation, in tumor cells [20]. Gallium can create complexes with proteins and ligands that typically bind iron, such as transferrin [19]. These interactions between gallium and proteins play a significant role in the potential development of gallium compounds as therapeutic agents in cancer [21,22]. Despite its similarities with iron, gallium distinguishes itself by not undergoing a trivalent to divalent state transition and being unable to engage in redox reactions.

To achieve a synergic effect, bioactive materials containing metals can be loaded externally with a pharmacologically active molecule. The silica structure of the classical mesoporous silica (MS) obtained with a Sol-Gel EISA pathway can achieve only superficial loading and pore blocking [23,24], which is well known for mesoporous materials. The best choice to obtain filling of pores is to synthesize a large pore mesoporous silica structure (LPMS). Considering this purpose, the structure can be tuned to be able to host a higher amount of the therapeutic molecule and to shield it from the external environment [25]. 

## 2. Materials and Methods

### 2.1. General Procedures

The chemicals and solvents, acquired at the highest available purity grade, were employed without additional purification unless explicitly stated. Microwave-assisted hydrothermal treatments using FlexiWAVE (Milestone S.r.l., Sorisole, Italy) at 230 V were conducted to synthesize LPMSs.

### 2.2. Synthesis of LPMSs

The synthesis of LPMS was adapted from earlier studies [25]. The previous synthesis was modified to achieve the prefixed content in calcium and gallium due to the high solubility of salt used. For the reference synthesis, the amount of water was decreased to a third.

To produce LPMSs, the synthesis began with an acidic water solution (1.7% *w*/*w* HCl) containing the following components: tetraethyl orthosilicate (TEOS) (98%, Merck, Darmstadt, Germany), Pluronic^®^ surfactant F127 (>99.9%, Merck, Darmstadt, Germany), 1,2,3-Trimethylbenzene (TMB) (100%, Merck, Darmstadt, Germany), Ca(NO_3_)_2_·4H_2_O (>99.0%, Merck, Darmstadt, Germany), and Ga(NO_3_)_3_·9H_2_O (>99.9%, Thermo Fisher Scientific, Waltham, MA, USA). The solutions underwent hydrothermal treatment with microwaves, initially subjected to a thermal treatment at 80 °C for 6 h, followed by a second thermal treatment at 160 °C for 12 h. Microwave power was regulated by the instrument based on the solution temperature.

All syntheses conducted are reported in Table 1. Samples were named as LPMS_nMe, in which n is the %mol of the Metal (Me) with respect to SiO_2_, and Me is replaced with the chemical symbol of the Metal element added to the silica structure. The amount of Ca(NO_3_)_2_·4H_2_O and Ga(NO_3_)_2_·9H_2_O, reported in Table 1, were optimized to obtain in the synthesized glasses the desired compositions (LPMS_5Ca refers to 100SiO_2_-5CaO, LPMS_5Ga refers to 100SiO_2_-2.5Ga_2_O_3_, and LPMS_5Ca5Ga refers to 100SiO_2_-5CaO-2.5Ga_2_O_3_ in %mol). The final glass composition was checked using XRF analysis.

Following the thermal treatment, the synthesis solutions underwent filtration, and the obtained solids were dried at 60 °C overnight. Subsequently, the solids were subjected to calcination at a rate of 1.5 °C/min, reaching 700 °C for 3 h under an air atmosphere. This process aimed to eliminate the surfactant and stabilize the resulting mesoporous glasses. After quenching in air, the LPMSs were carefully milled in an agate mortar and sieved to achieve a mean dimension below 355 µm.

### 2.3. Load Tests and Release Tests

Load and release tests were performed to compare LPMSs’ behavior when contained or not one or more metals. Tests were conducted as in our previous article [25].

To assess the quantity of nisin encapsulated within the structures and the amount released during immersion in simulated body fluid (SBF), UV-VIS spectra were obtained. The cumulative release percentage (%) of nisin was determined in order to control the residual % of nisin not able to be released by the porous materials. UV-visible spectra were recorded using a JASKO V-570 UV/Vis/NIR spectrophotometer (JASCO Europe S.r.l., Cremella, Italy) at 298 K within the 250–400 nm spectral range, utilizing quartz cells with a 1 cm optical path. The quantification of nisin was carried out at its peak absorbance, which occurred at 277 nm.

To study the kinetic mechanism of nisin release, we applied the equation (Korsmeyer–Peppas): M_t_/M_∞_ = k − t^n^, where M_t_/M_∞_ corresponds to the fraction of drug released at time t, k is the diffusion rate constant, and n is the release exponent that indicates the mechanism of release [26]. Nisin release was performed for 4 days; however, the model must be applied to the linear part of the release when the release has a significant value.

If n, obtained from the linearization of the plot of ln (% nisin release) vs. ln (time) is <0.5, the release is under partial diffusion control, while if it is equal to 0.5, the release is under diffusion control. If n = 1, the release mechanism is controlled by the erosion of the delivery system.

### 2.4. Gallium, Calcium, Silica, and Phosphorous Release in SBF

Metal ions released in SBF were evaluated using ICP-OES. Firstly, 100 mg of solid dried samples were put in contact with 20 mL of SBF. After 1, 3, 7, or 14 days, supernatants were collected, properly diluted, and acidified, and then the quantification of the amount of gallium, calcium, silica, and phosphorus released was performed. For each time frame, a different solution was prepared according to what was found in the literature [27,28,29,30]; for the inorganic ion leaching, it is important to detect the maximum concentration obtainable near the material implanted in order to obtain the biological effect (i.e., antitumoral effect). For the analyses, an ICP-OES DA 4500 Perkin Elmer, Perkin Elmer Optima 4200 DV (Perkin Elmer, Milan, Italy) was used.

### 2.5. Bioactivity Evaluation after Soaking with SBF

To evaluate the bioactivity of samples, a specific amount of powdery sample was kept in contact with SBF for 1-3-7-14 days. The formation of a hydroxyapatite (HA) layer was evaluated through IR spectra, checking the characteristic IR peaks. IR spectra were recorded in the 4000–400 cm^−1^ spectral range using an FTIR VERTEX 70 (Bruker, Milan, Italy). In addition, XRDP was used to confirm the presence of a layer of HA on the surface of the solid after contact with SBF. Diffraction spectra were acquired in the range of 10–60° in 2θ, Cu Kα of 1.5418 Å, 40 kV e 40 mA, maximum power 2.2 kW using X’Pert PRO (Panalytical, Malvern, UK).

### 2.6. Physical–Chemical Characterization of Powders

#### 2.6.1. X-ray Fluorescence Spectroscopy (XRF)

The determination and the control of the composition of powdery samples synthesized were conducted through XRF analysis using an X Philips PW 1480 (Panalytical, Lissone, Italy).

#### 2.6.2. Scanning Electron Microscopy (SEM)

The morphology of the pristine powdery samples was examined to verify the existence of a substantial porous structure consistent with previous findings. Additionally, SEM images were captured for both loaded powdery samples and those subjected to SBF immersion, employing a Nova NanoSEM 450 microscope (FEI Company, Milan, Italy) operating at 15 kV.

#### 2.6.3. Textural Properties

To assess the surface characteristics of LPMSs, surface areas were determined through N_2_ adsorption/desorption isotherms conducted at approximately 77 K using a TriStar II 3020 Micromeritics instrument (Alfatest S.r.l., Rome, Italy). The adsorption data were analyzed using the standard Brunauer, Emmet, and Teller (BET) method [31] to determine the specific surface area (SSABET). The pore size distribution was determined using the Barret-Joyner-Halenda (BJH) method [32] from the adsorption branch of the isotherm. Furthermore, the Total Pore Area and the Intrusion Volume were determined using a mercury porosimeter. This was accomplished with an AutoPore IV 9500 instrument (Micrometrics Instrument Corporation, Alfatest S.r.l., Rome, Italy) operating at a mercury filling pressure of 1.51 psi.

#### 2.6.4. Elemental Analysis (EA)

Elemental analysis (EA) was conducted to validate the information obtained from UV-VIS spectra and quantify the quantity of nisin encapsulated within the structure. Thermo Scientific™ FLASH 2000 CHNS analyzer (Thermo Fischer Scientific Inc., Milan, Italy) was employed for this purpose. A process blank was prepared and analyzed alongside all samples, with the results duly taken into consideration.

#### 2.6.5. Thermogravimetric Analysis (TG-DTA)

To affirm the stability of nisin within the structures and assess specific interactions between nisin and the silica surface, thermal gravimetric (TG) analyses were conducted. The analyses were carried out using a Seiko SSC 5200 instrument (Seiko Instrument Inc., Chiba, Japan) over a temperature range spanning from 25 °C to 650 °C, employing a heating rate of 1 °C/min.

#### 2.6.6. Confocal Laser Scanning Microscopy (CLSM)

A confocal microscope was employed to assess the presence of nisin on the surface of powdery samples loaded with the compound. Nisin fluorescence was generated by utilizing an excitation wavelength of 405 nm, and the emitted light was recorded within the 470–580 nm range. This imaging was performed using a Leica TCS SP8 microscope (Leica Biosystems, Milan, Italy).

## 3. Results and Discussion

### 3.1. LPMSs’ Morphology Characterization with SEM

The structures of LPMSs prepared as reported above were checked with SEM-FEG to control the correct morphology due to the modification of composition and the change in water amount on the synthesis compared with the previous work [25]. 

With changing composition and water amount, structures always present large pores (see Figure 1). To compare the result obtained with LPMSs, a reference structure used MS, synthesized via the Sol-Gel EISA process [22] with the same composition in calcium and gallium. MS structures were named LPMSs, obtaining MS_5Ga, MS_5Ca, and MS_5Ca5Ga (see Figure 2 to check the surface morphology of MS).

### 3.2. SSA_BET_ and Hg Porosimeter Characterization

Through SSA_BET_ and Hg porosimetry, it is possible to demonstrate the difference in textural properties (i.e., pore dimensions) between LPMSs and MSs (reported in Appendix A and Figure 3, respectively). 

As it is possible to read from Appendix A, no significant difference in SSA_BET_ is detected; the SSA for LPMS samples range is 283–324 m^2^/g, while for the MS sample, the range falls in the 252–417 m^2^/g range. Moreover, these results are in line with that reported in the literature [8]; the SSA reported of these bioactive glasses is around 400 m^2^/g [33]. Also, the amount of N_2_ adsorbed/desorbed could be considered to study the difference in pore characteristics (Appendix A); in fact, it is noticeable that LPMS showed a pore volume five times higher compared with MSs when considering the same amount of sample, thanks to their structure rich in cavities and large pores. In fact, classical mesoporous bioactive glasses reach a maximum value of 0.4 cm^3^/g, while for our LPMS bioactive glasses, the value is higher, in the 1.56–1.77 cm^3^/g range. 

This characteristic was confirmed by the Total Pore Areas (see Appendix A) detected with Hg intrusion. The Total Pore Area obtained for LPMS falls in the 392–344 m^2^/g range, while the range of the MS sample is lower (179–53 m^2^/g). This is due to the Hg penetration method; this method can be applied if the pores are larger than 3.5 nm (this limit depends on the properties of sample compressibility) [34]. In MS samples, the number of pores with a diameter less than 3.5 nm is very high, so Hg intrusion is not able to fill all these small pores; this phenomenon does not occur with N_2_.

As it is possible to see from Figure 1, all LPMSs show mesopores in the ranges 20–60 nm and 200–600 nm, and all MSs only show mesopores in the range 2–5 nm. LPMS pores, according to previous works [35,36], could be classified, respectively, as Super-Nanopores and Super-Micropores. From Appendix A, it is also noticeable that all structures synthesized present Type IV pores [37,38,39].

### 3.3. Nisin Load and Release Tests 

The effectiveness of loading a molecule into a structure can be quantified through various metrics, including loading efficiency percentage (LE%) [40], loading percentage (Loading%) [41], or loading capacity percentage (LC%) [42,43]. These parameters are different ways to estimate the amount of active molecule loaded into a structure, with respect to the total amount of molecule kept in contact with silica structure or with respect to the initial silica amount.

Elemental analysis (EA) was conducted on solids collected postloading and subsequently dried overnight. Table 2 presents the values for LE%, LC, Loading%, and Loading% calculated through EA. The samples are denoted as “sample_name_x”, where x signifies the concentration of nisin (in mg/mL) in the loading solution.

From Table 2, it is possible to notice that MSs give the same results in terms of efficiency of loading only when loaded with a solution with a double concentration of nisin. This behavior is justifiable considering the filling of pores of LPMS, which brings it to higher loading capacity, a phenomenon that does not occur for MSs when only pore blocking takes place [24,41]. In addition, LC% measured for MSs kept in contact with 5mg/mL nisin solution is three times lower compared with the corresponding LPMSs structures, results according to what is reported in Appendix A; in fact, also in this case, this behavior can be explained considering the high difference in pore volume and pore size that there is between LPMSs and MSs.

Concerning release tests presented in Figure 4, LPMSs show better behavior compared with MSs, also considering the release time of the same amount of nisin. The same amount of nisin is released in extremely longer times for LPMSs instead of corresponding MSs, from two to eighty times longer, confirming a more controlled and prolonged release compared with classical structures. 

Data presented in Figure 4 are coherent with ones obtained in our previous work [25]; for LPMSs composed only from SiO_2_, a release in 4 days was registered, a doubled time if compared to the 2 days obtained for MSs. 

In addition, for these structures, an influence of the synergic effect of the presence of Ca and Ga in the structure is evident. When both metals are present together in the structure, a general decrease in the releasing time is registered.

The n values for the Korsmeyer-Peppas model were reported in Appendix A. The n values are always <1; however, for all LPMS and MS_5Ca5Ga samples, the n value is around 0.2, suggesting that the nisin release is under partial diffusion control, while for MS_5Ca and MS_5Ga samples, the n value is 0.9, confirming a different type of release controlled mainly by the erosion.

Comparing our results with some others reported in the literature for similar structures [43,44], it is noticeable that a longer release could be obtained (4 days for LPMSs compared with 50 h of MSs) and a total release could be achieved (instead of 60% obtained from Hosseinpour et al. [43]). 

To confirm the good mechanical properties of LPMSs, SEM-FEG images of samples loaded and after the entire release have been acquired (Appendix A). The morphology of samples after the nisin release is characteristic of porous structure, confirming the presence of pores in the samples after the drug release. We can conclude that the collapse of porous structure during the release does not occur in our LPMS in contrast to that often found in the literature [45], where the collapsing of pore structure during the drug release tests causes an incomplete release of drugs. As a confirmation of the presence of nisin on the surface of the materials and as a confirmation of its complete release, a confocal laser scanning microscope was used to acquire images of samples (see Figure 5).

### 3.4. Thermogravimetric Analysis (TG-DTA)

To evaluate the influence on the interactions between nisin and the surface of silica when metals are present on the surface of the compounds, a TG_DTA analysis was performed (Figure 6). In the DTG graph, the first peak is related to the pyrolysis of nisin, and the second peak is related to the decomposition of the structure of nisin [46]. From derivative graphs obtained from TG (DTG), it is possible to notice that when gallium is preset on the surface, some interactions are promoted because the second peak is shifted to the right, and a third peak is present. The same behavior is registered both for structures with only Ga and in structures with Ca and Ga. This behavior could also be considered to explain the longer releasing time for a structure containing Ca and Ga (Figure 4). 

### 3.5. Gallium, Calcium, Silica, and Phosphorous Release in SBF 

To evaluate the antitumoral potential of the compound synthesized, gallium release in SBF was examined. Contemporarily, to confirm the formation of a layer of HA on the surface of the powdery compound, control of the amount of Ca and P in SBF was performed (Figure 7).

Regarding the quantity of Ga released (Figure 7), LPMSs release an amount four times higher than MSs in a time range of 14 days. The oscillation in the amount of Ga at the beginning of the release could be attributed to the formation of gallium phosphates, oscillation not detectable for LPMS_5Ga probably due to its extremely high release. The LPMS_5Ga and LPMS_5Ca5Ga samples showed in SBF the maximum release of Ga^3+^ ions; moreover, the maximum concentration reached in SBF was around 8 ppm. These values are lower than reported for the toxicity limit in blood plasma (14 ppm) [47,48]. With these Ga^3+^ ions’ release, it can be assumed an antibacterial activity; S. Pourshahrestani et al. [49] demonstrate that with a release of only 0.3 ppm of Ga^3+^, an inhibition of 80% and 100% is reached, respectively, for *E. coli* and *S. aurus*. In addition, L. Antunes et al. [50] demonstrate that a release of 2.5 ppm of Ga^3+^ is sufficient to inhibit about 90% of the growth of *A. baumannii*. Moreover, several studies [51,52,53] state that the antitumoral activity of Ga^3+^ is shown in a concentration between 100 µM and 1 mM according to the different tumoral cells, resulting in accord to that obtained in this work bringing to the consideration that our compound could show a potential antitumoral effect. 

Calcium, in shorter times, is heavily released from LPMS_5Ca. This behavior causes the formation of a small amount of calcite, detectable also through X-ray powder diffraction. The formation of calcite could be explained by considering that it kinetically favors the formation of calcite and thermodynamically favors the formation of HA, which is detectable after 14 days.

Contemporary to the increase in calcium, the amount of phosphate for the samples that contain Ca decreases due to the formation of calcium phosphate. To confirm the formation of HA, X-ray powder diffraction on powders and FT-IR spectra were acquired. 

Concerning Si, it has been controlled so that its amount always stays near 60 ppm, its solubility in SBF [54,55].

### 3.6. Bioactivity Evaluation after Soaking with SBF

A bioactivity evaluation after soaking with SBF was performed, registering FT-IR spectra of samples taken in contact for a specific time-lapse with SBF (1, 3, 7, or 14 days). All samples containing calcium presented HA characteristic peaks at 560 cm^−1^ and 605 cm^−1^ [56,57] (Figure 8, for the entire spectra, see Appendix A). 

As a further confirmation of the presence of Hydroxyapatite (HA) on the surface of powdery samples after soaking with SBF, X-ray powder diffraction of samples soaked for 14 days has been acquired. Looking at Figure 9, it is possible to notice that all samples containing Ca, both LPMSs and MSs, show HA characteristic peaks [58,59] at 26° and 32° in °2θ, confirming the presence of HA. In addition, for LPMSs containing Ca, it is possible to see a peak related to the formation of Calcite (CA) (29° in °2θ) due to the high release of Ca^2+^ ions in the first hours of soaking [60]. This can be explained by considering the kinetically favored formation of calcite during the first hour, which is replaced during the time from the formation of HA, which is thermodynamically more stable [60,61].

The morphology of the surface was observed in order to confirm the presence of HA after soaking with SBF; an SEM-EDS analysis of samples soaked for 14 days was performed, and the results are reported in Figure 10.

After the soaking on the sample surfaces, it is possible to detect the formation of a new phase, but some differences can be noted. In the LPMS_5Ca sample, the new layer formed is uniform and tends to cover the porous structure of the material, and the EDS analysis shows a great increment of Ca and P on the surface. Ca/P is 1.80, near to the theoretical Ca/P ratio in the HA crystal phase (1.67). The slightly high Ca/P ratio may be due to the presence of the calcite phase. In the case of LPMS_5Ca5Ga (Figure 10c), it is possible to note the formation of a new layer, but this new layer does not completely cover the surface, while in Figure 10b, it is possible to find only small particles of new formation on the surface; in fact, in this sample, the increment of P and Ca on the glass surface is lower compared with other LMPS samples.

For MS samples, it is not possible to see a new layer formed on the glass surface after SBF soaking; however, in the case of MS_5Ca (Figure 10d), clearly evident is the formation of a new phase where Si, Ca, and P are simultaneously present, and the Ca/P ratio is 1.60, very close to the theoretical value for the HA crystal phase.

## 4. Conclusions, Limitations and Prospects

Based on the obtained results, it can be inferred that the previously established synthesis pathway was successfully applied to produce Ca and Ga-containing LPMSs while maintaining consistent characteristics in terms of pore distribution. In this investigation, LPMSs exhibited enhanced features in loading and release tests compared to MS, reaching a doubled releasing time and a doubled loading efficiency, demonstrating controlled and prolonged release for 48 h, as well as improved mechanical resilience in a physiological environment.

In terms of bioactivity, Ca-containing LPMSs demonstrated bioactivity similar to that already studied and well-known for MS. Notably, an enhancement in ion release was achieved with LPMSs, releasing a fourfold higher amount of Ga (8 ppm), known for its anticancer properties. This concentration guarantees antibacterial and antitumoral activity.

In summary, LPMSs emerge as promising candidates for achieving high loading and controlled drug release of sizable pharmaceutical molecules, owing to their structured pore system and mechanical durability. Their application in investigating the mobility of large organic molecules in a confined state could be contemplated. Additionally, the inclusion of phosphorous may be considered to impart bioactivity to LPMSs containing only Ga. Lastly, exploring new substantial pharmaceutical molecules for further studies is warranted.

Nevertheless, a potential vulnerability of the synthesized structures may lie in the potential obstruction of superficial pores by substances encountered in vivo, such as cells.

## Figures and Tables

**Figure 1 materials-17-00373-f001:**
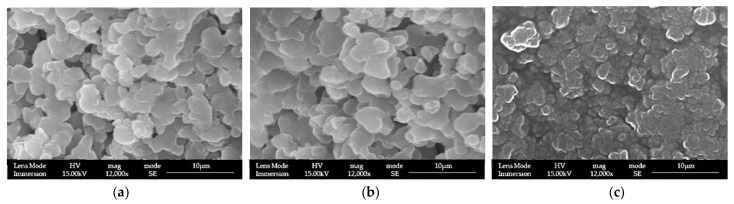
SEM-FEG images of (**a**) LPMS_5Ga, (**b**) LPMS_5Ca, and (**c**) LPMS_5Ca5Ga.

**Figure 2 materials-17-00373-f002:**
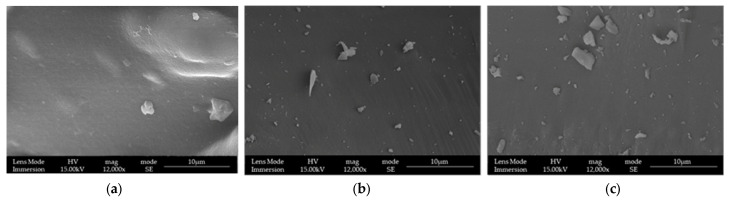
SEM-FEG images of (**a**) MS_5Ga, (**b**) MS_5Ca, and (**c**) MS_5Ca5Ga.

**Figure 3 materials-17-00373-f003:**
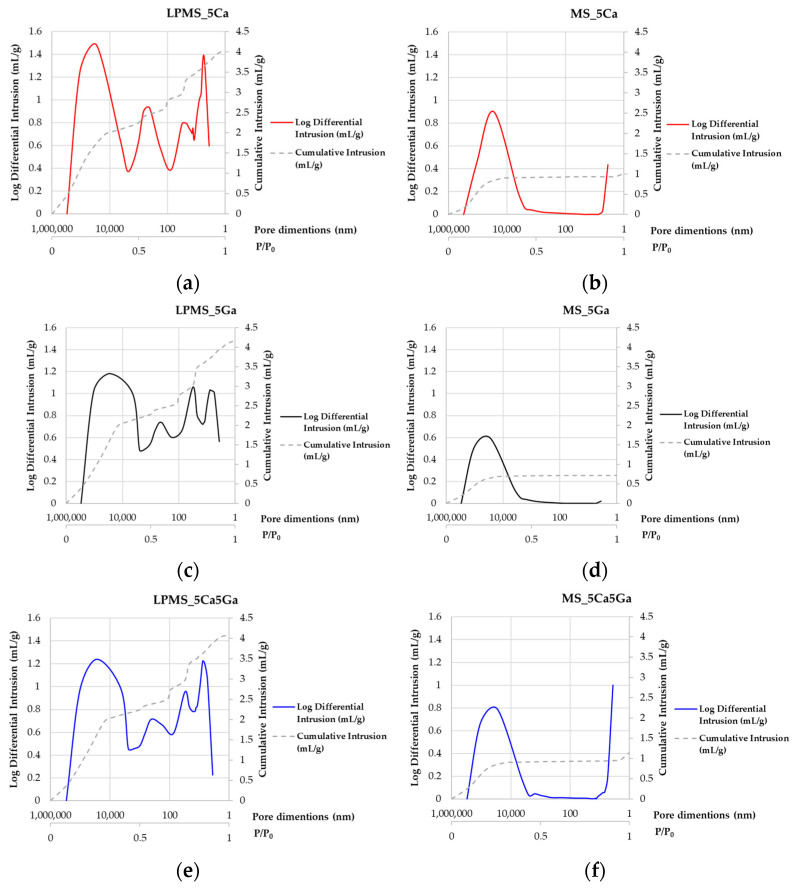
Log differential intrusion and cumulative intrusion evaluated with Hg porosimeter of all samples. In order: (**a**) LPMS_5Ca, (**b**) MS_5Ca, (**c**) LPMS_5Ga, (**d**) MS_5Ga, (**e**) LPMS_5Ca5Ga, and (**f**) MS_5Ca5Ga.

**Figure 4 materials-17-00373-f004:**
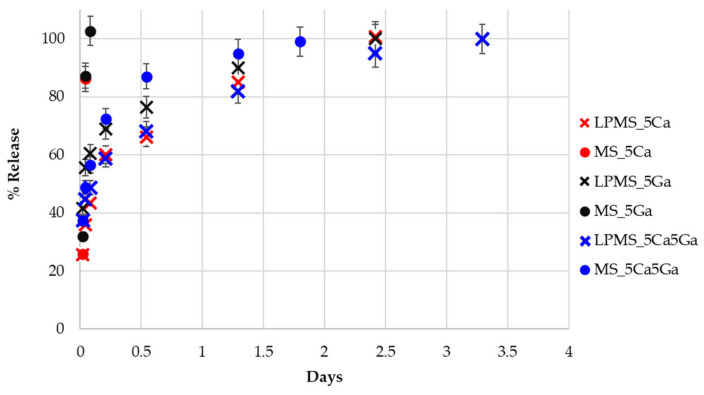
Release tests of LPMSs and MSs. The %Release determined with UV-Vis analysis plotted as a function of time.

**Figure 5 materials-17-00373-f005:**
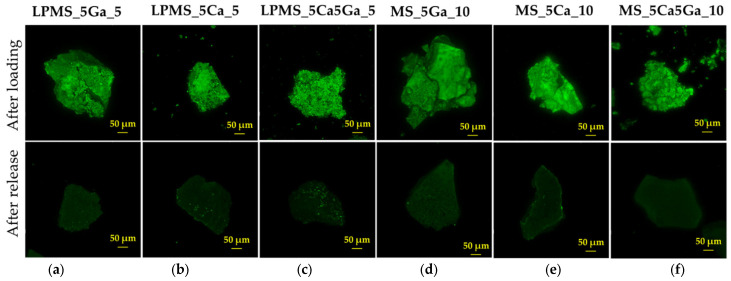
CLSM images of (**a**) LPMS_5Ga_5, (**b**) LPMS_5Ca_5, (**c**) LPMS_5Ca5Ga_5, (**d**) MS_5Ga_10, (**e**) MS_5Ca_10, and (**f**) MS_5Ca5Ga_10. In the first row are reported images of loaded structures, and in the second row are reported structures after release.

**Figure 6 materials-17-00373-f006:**
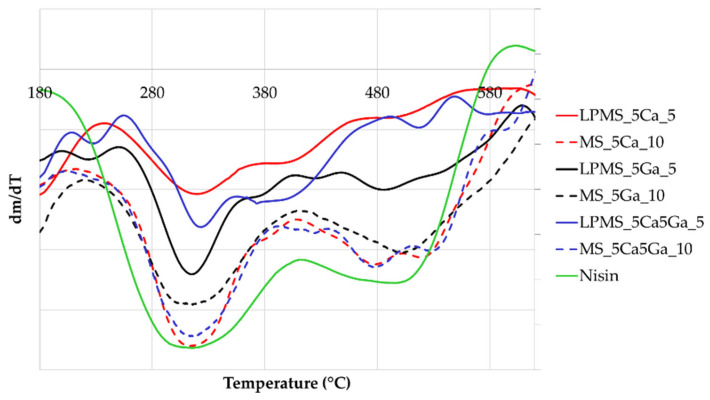
DTG, derivative graph obtained from TG curves of all samples, and a sample of pure nisin.

**Figure 7 materials-17-00373-f007:**
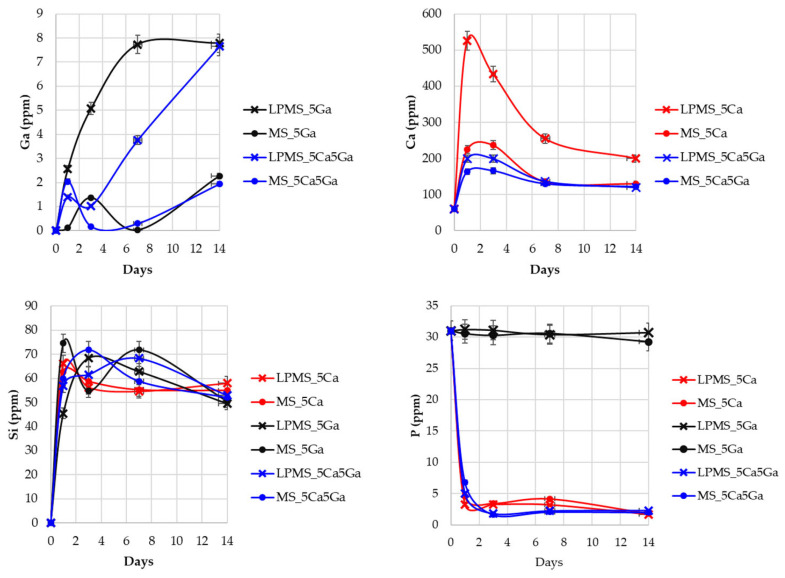
Releases of Ga, Ca, Si, and P in SBF, acquired through ICP-OES in a time interval of 14 days.

**Figure 8 materials-17-00373-f008:**
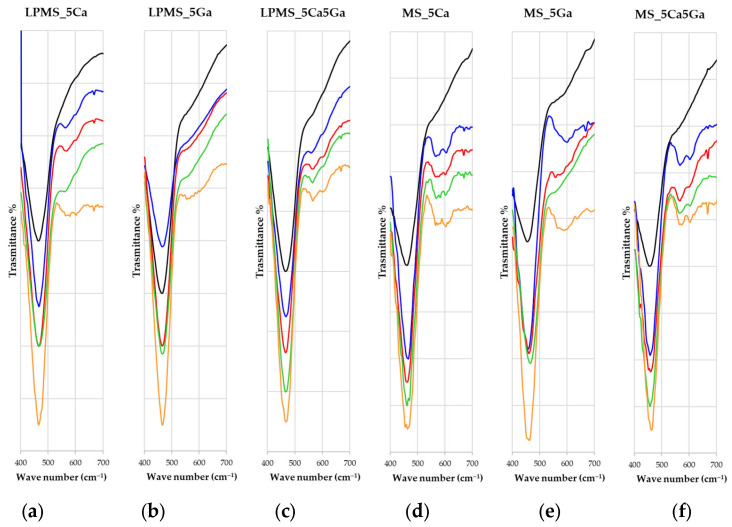
FT-IR spectra of all samples acquired at 1, 3, 7, and 14 days for samples prepared as KBr pads. In order: (**a**) LPMS_5Ca, (**b**) LPMS_5Ga, (**c**) LPMS_5Ca5Ga, (**d**) MS_5Ca, (**e**) MS_5Ga, and (**f**) MS_5Ca5Ga.

**Figure 9 materials-17-00373-f009:**
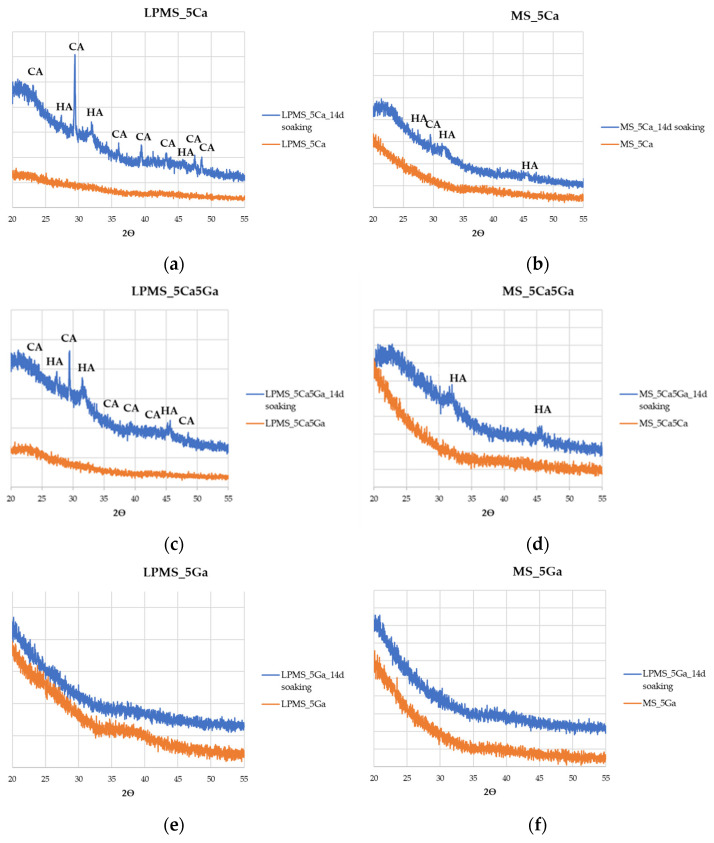
XRPD was recorded for as-synthesized samples (orange) and samples after 14 days of soaking in SBF. In order: (**a**) LPMS_5Ca, (**b**) MS_5Ca, (**c**) LPMS_5Ca5Ga, (**d**) MS_5Ca5Ga, (**e**) LPMS_5Ga, and (**f**) MS_5Ga.

**Figure 10 materials-17-00373-f010:**
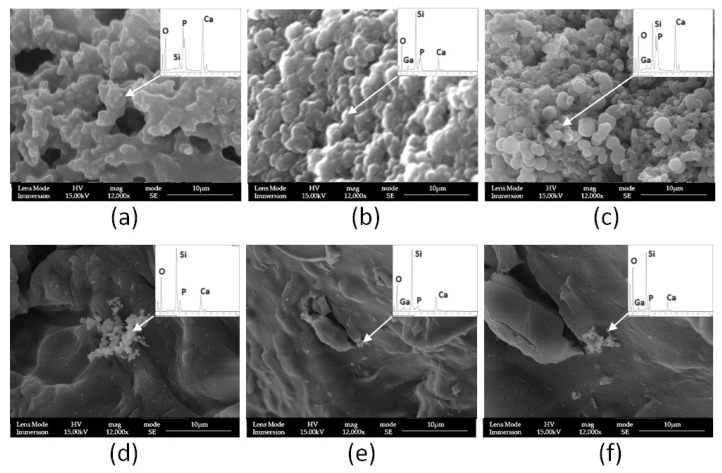
SEM-EDS micrographs for samples after 14 days of soaking in SBF. In order: (**a**) LPMS_5Ca, (**b**) LPMS_5Ga, (**c**) LPMS_5Ca5Ga, (**d**) MS_5Ca, (**e**) MS_5Ga, and (**f**) MS_5Ca5Ga (inset showed the EDS spectra).

**Table 1 materials-17-00373-t001:** Synthesis conducted.

Sample	TEOS (g)	Pluronic^®^ F127 (g)	TMB (mL)	HCl (1.7%) *w*/*w* (mL)	Ca(NO_3_)_2_·4H_2_O (g)	Ga(NO_3_)_3_·9H_2_O (g)
LPMS_5Ca	2.10	4.20	0.55	40	0.90	\
LPMS_5Ga	2.10	4.20	0.55	40	\	0.93
LPMS_5Ca5Ga	2.10	4.20	0.55	40	0.90	1.05

**Table 2 materials-17-00373-t002:** Comparison between LE%, LC%, loading%, and loading% calculated with EA of all samples studied.

Sample	LE%	LC%	Loading%	Loading% with EA
LPMS_5Ca_5	26.4	8.0	7.4	8.9
LPMS_5Ga_5	27.6	8.4	7.7	9.4
LPMS_5Ca5Ga_5	23.4	7.1	6.6	7.3
MS_5Ca_5	8.1	2.8	2.7	3.6
MS_5Ga_5	12.4	4.1	3.9	3.8
MS_5Ca5Ga_5	13.1	4.3	5.1	4.8
MS_5Ca_10	29.8	9.6	8.7	9.4
MS_5Ga_10	20.5	6.4	6.0	7.5
MS_5Ca5Ga_10	17.3	10.3	9.3	9.0

## Data Availability

Additional data that support the findings of this study are available from the corresponding author.

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
