# Peer review of "Very Large Pore Mesoporous Bioactive Silicate Glasses: Comparison of Behavior toward Classical Mesoporous Bioactive Glasses in Terms of Drug Loading/Release and Bioactivity"

_materials, 2024, doi:10.3390/ma17020373_

Round 1
Reviewer 1 Report
Comments and Suggestions for Authors
FEG-SEM micrographs must be incorporated to the results, and not provided as a supplementary file.
The micrographs shown in the supplementary file are not acceptable for publication. They show strong charging effect.
Confocal results must also be inserted in the main document.
The authors did not mention bone-like apatite formation after incubation in SBF. These results must be presented and the micrographs of the samples after incubation in SBF must be discussed.

Comments on the Quality of English LanguageEnglish spelling review must be done.
Author Response
Reviewer 1
1) FEG-SEM micrographs must be incorporated to the results, and not provided as a supplementary file.
SEM-FEG images were removed from supplementary materials and added to the text (respectively Figure 1 and 2).
2) The micrographs shown in the supplementary file are not acceptable for publication. They show strong charging effect.
New imagine were reported with a better quality (new version of Figure 1, Figure 2 in the paper and Figure S3 in the Supplementary materials file).
3) Confocal results must also be inserted in the main document.
Confocal images were removed from supplementary materials and added to the text (see Figure 5).
4) The authors did not mention bone-like apatite formation after incubation in SBF. These results must be presented and the micrographs of the samples after incubation in SBF must be discussed.
In the section 3.6 was added the SEM-EDS analysis of sample surfaces after SBF in order to check the HA formation. A new Figure (Figure 10) and comments was added in the new version of the paper.
Reviewer 2 Report
Comments and Suggestions for Authors
The article "Very large pores mesoporous bioactive silicate glasses: comparison behavior towards classical mesoporous bioactive glasses in term of drug loading/release and bioactivity" contains an interesting research and deserves to be published, after its correction.
- Abstract contains an introduction that should be part of the text rather than the abstract. That is, it should be reduced.
-The article contains several typos, such as:
· row 9 Synthetizing... instead of Synthesizing...
· row 11 "Large Porse Mesoporous Silica (LPMS)" instead of "Large Pores Mesoporous Silica (LPMS)"
· -row 75 is 67Ga, and row 76 is 67Ga
· -row 310 is "E. Coli and S. Aurus" instead of "E. Coli and S. Aureus"
· -row 312 A. Baumannii instead of A. Baumannii
· row 322 "Contemporarily to the increase of Calcium..."
· row 365 "for achieving high loading and controlled drug release..."
- some figures from the supplementary material should perhaps appear in the article. As, for example, Fig. S4.
-all the literature in the text is not cited according to the journal's requirements
- Conclusions must be enriched with some data from the text
- The bibliography is not written according to the requirements of the journal

Comments on the Quality of English LanguageEnglish needs correction. The entire text needs to be revised.
Author Response
Reviewer 2
- Abstract contains an introduction that should be part of the text rather than the abstract. That is, it should be reduced.
The Abstract was reduced, and the first part has been added to the introduction and the proper citation has been added.
- The article contains several typos, such as:…
Typos have been corrected.
- Some figures from the supplementary material should perhaps appear in the article. As, for example, Fig. S4.
Figure S4 (and Figures S1 and S2) have been added to the article and have been removed from Supplementary Materials
- All the literature in the text is not cited according to the journal's requirements - The bibliography is not written according to the requirements of the journal.
The literature has been corrected during the first revision.
- Conclusions must be enriched with some data from the text.
Conclusions have been implemented with some data, in particular we added the maximum time achieved for the Nisin release and highest value of Ga3+ concentration obtained after the soaking in SBF.
Reviewer 3 Report
Comments and Suggestions for Authors
In this work, the authors report on the synthesis and systematic characterization of Large Pores Mesoporous Silica (LPMS) hosting therapeutic ions Ca(II) and Ga(III). I have a major concern about the release studies and I also believe that this manuscript can be improved in clarity.
The authors say: “After 1,3,7 or 14 days supernatants were collected, properly diluted, acidified and then the quantification of the amount of Gallium, Calcium, Silica and Phosphorus released was performed.” Are release studies carried out in a way that sink condition, which is relevant to biological systems, could be approximately assumed. Are separated identical samples with supernatant characterized at different times or is the supernatant replaced with fresh SBF between after a measurement? I think the authors’s experiments fall within the former case but they would not necessarily correspond to sink condition. A cumulative release would then be a monotonic function of time and would not exhibit the oscillatory-like behavior shown in Fig. 4. The authors could substantiate their approach by citing established work in their field from other research groups.
What type of time dependence the experimental curves in Fig. 2 exhibit? Is it diffusion-like?
Minor concerns:
In Abstract: “Porse” should be Pores. Ca and Ga without extra symbols indicating charges are not ions. There are abbreviations such as ICP-OES and EISA that are not specified and may be not clear to a reader. In general, the abstract should be simplified and it seems to me that writing style can be improved for clarity purpose. Finally, abstract mentions peptide and proteins but this work is on ions.
The authors say: “The effectiveness of loading a molecule into a structure can be quantified through 237 various metrics, including loading efficiency percentage (LE%) [35], loading percentage 238 (Loading%) [36], or loading capacity percentage (LC%) [37], [38].” I know that these loading quantities are cited but I suggest that the authors defines them in one or two text lines.
The authors start a paragraph with: “Concerning releasing tests, presented in Figure 2, LPMSs show a better behaviour also considering the time of release the same amount of Nisin. In fact, the same amount of Nisin is released in extremely longer times for LPMSs, from twice to eighty times longer, confirming a controlled and prolonged release.” These statements are vague. Better behavior than what? “the time of release the same amount of Nisin” is grammatically incorrect. “the same amount of Nisin is released in extremely longer times” than what? Finally, how do you quantitatively define “controlled and prolonged release”?
Comments on the Quality of English Languagesee my comments above
Author Response
Reviewer 3
- The authors say: “After 1,3,7 or 14 days supernatants were collected, properly diluted, acidified and then the quantification of the amount of Gallium, Calcium, Silica and Phosphorus released was performed.” Are release studies carried out in a way that sink condition, which is relevant to biological systems, could be approximately assumed. Are separated identical samples with supernatant characterized at different times or is the supernatant replaced with fresh SBF between after a measurement? I think the authors’s experiments fall within the former case but they would not necessarily correspond to sink condition. A cumulative release would then be a monotonic function of time and would not exhibit the oscillatory-like behavior shown in Fig. 4. The authors could substantiate their approach by citing established work in their field from other research groups.
For each day a different solution was prepared. A cumulative approach was used only for the study of Nisin release in SBF. In particular, the cumulative release percentage (%) of Nisin was determined in order to control the residual % of Nisin not able to release by the porous materials.
Regarding the method to obtain the inorganic ions release some articles have been added (see section 2.4) as citations for supporting the chosen approach. In fact, for the inorganic ions leaching it is important to detect the maximum concentration obtainable near the implant in order to obtain biological effect (i.e. antitumoral effect).
- What type of time dependence the experimental curves in Fig. 2 exhibit? Is it diffusion-like?
In the new version of the paper we studied the kinetic mechanism of Nisin release using the Korsmeyer–Peppas equation. The n values found for the Korsmeyer-Peppas model were always <1, however for all LPMS and MS_5Ca5Ga samples the n value is around 0.2 suggesting that the nisin release is under partial diffusion control.
- In Abstract: “Porse” should be Pores. Ca and Ga without extra symbols indicating charges are not ions. There are abbreviations such as ICP-OES and EISA that are not specified and may be not clear to a reader. In general, the abstract should be simplified and it seems to me that writing style can be improved for clarity purpose. Finally, abstract mentions peptide and proteins but this work is on ions
The abstract has been improved, adding the meaning of acronyms and reformulating some sentences. Peptides and proteins have been cited because of the study of loadings and releases using Nisin (a polycyclic peptide).
- The authors say: “The effectiveness of loading a molecule into a structure can be quantified through 237 various metrics, including loading efficiency percentage (LE%) [35], loading percentage 238 (Loading%) [36], or loading capacity percentage (LC%) [37], [38].” I know that these loading quantities are cited but I suggest that the authors defines them in one or two text lines.
LE%, loading% and LC% have been defined and explained in lines 265-274.
- The authors start a paragraph with: “Concerning releasing tests, presented in Figure 2, LPMSs show a better behaviour also considering the time of release the same amount of Nisin. In fact, the same amount of Nisin is released in extremely longer times for LPMSs, from twice to eighty times longer, confirming a controlled and prolonged release.” These statements are vague. Better behavior than what? “the time of release the same amount of Nisin” is grammatically incorrect. “the same amount of Nisin is released in extremely longer times” than what? Finally, how do you quantitatively define “controlled and prolonged release”?
This section (3.3) has been improved.